# Recent Progress of Miniature MEMS Pressure Sensors

**DOI:** 10.3390/mi11010056

**Published:** 2020-01-01

**Authors:** Peishuai Song, Zhe Ma, Jing Ma, Liangliang Yang, Jiangtao Wei, Yongmei Zhao, Mingliang Zhang, Fuhua Yang, Xiaodong Wang

**Affiliations:** 1Engineering Research Center for Semiconductor Integrated Technology, Institute of Semiconductors, Chinese Academy of Sciences, Beijing 100083, China; pssong@semi.ac.cn (P.S.); mazhe@semi.ac.cn (Z.M.); majing@semi.ac.cn (J.M.); yangliangliang@semi.ac.cn (L.Y.); weijt@semi.ac.cn (J.W.); ymzhao@semi.ac.cn (Y.Z.); Zhangml@semi.ac.cn (M.Z.); fhyang@semi.ac.cn (F.Y.); 2The School of microelectronics & Center of Materials Science and Optoelectronics Engineering, University of Chinese Academy of Sciences, Beijing 100049, China; 3The School of Electronic, Electrical and Communication Engineering, University of Chinese Academy of Sciences, Beijing 100190, China; 4Beijing Engineering Research Center of Semiconductor Micro-Nano Integrated Technology, Beijing 100083, China; 5Beijing Academy of Quantum Information Science, Beijing 100193, China

**Keywords:** pressure sensor, MEMS, graphene, piezoresistive, resonant pressure sensor, capacitive pressure sensor, optical fiber pressure sensor, piezoelectric pressure sensor, implantable pressure sensor, blood pressure, intraocular pressure, intracranial pressure

## Abstract

Miniature Microelectromechanical Systems (MEMS) pressure sensors possess various merits, such as low power consumption, being lightweight, having a small volume, accurate measurement in a space-limited region, low cost, little influence on the objects being detected. Accurate blood pressure has been frequently required for medical diagnosis. Miniature pressure sensors could directly measure the blood pressure and fluctuation in blood vessels with an inner diameter from 200 to 1000 μm. Glaucoma is a group of eye diseases usually resulting from abnormal intraocular pressure. The implantable pressure sensor for real-time inspection would keep the disease from worsening; meanwhile, these small devices could alleviate the discomfort of patients. In addition to medical applications, miniature pressure sensors have also been used in the aerospace, industrial, and consumer electronics fields. To clearly illustrate the “miniature size”, this paper focuses on miniature pressure sensors with an overall size of less than 2 mm × 2 mm or a pressure sensitive diaphragm area of less than 1 mm × 1 mm. In this paper, firstly, the working principles of several types of pressure sensors are briefly introduced. Secondly, the miniaturization with the development of the semiconductor processing technology is discussed. Thirdly, the sizes, performances, manufacturing processes, structures, and materials of small pressure sensors used in the different fields are explained in detail, especially in the medical field. Fourthly, problems encountered in the miniaturization of miniature pressure sensors are analyzed and possible solutions proposed. Finally, the probable development directions of miniature pressure sensors in the future are discussed.

## 1. Introduction

Pressure is one of the basic physical parameters that is tightly associated with life and production. Many approaches have been developed to measure pressure [1,2,3]. The pressure sensor is a device that could perceive a pressure signal and convert the pressure signal into an output electric signal according to certain mechanisms. Usually, it consists of pressure elements and a signal processing unit. According to the working principle, pressure sensors can be divided into piezoresistive, capacitive, optical fiber, resonant, and piezoelectric types [4]. The pressure sensors mentioned in this paper are all included in the five above working mechanisms. These principles will be briefly explained below.

For piezoresistive sensors, the sensitive diaphragms above the cavities are subjected pressure and generated stress. Figure 1a shows a typical structure of a piezoresistive absolute pressure sensor. The cavity was vacuum sealed by bonding the substrates together, and the silicon film was subject to pressure changes [5,6]. The designed piezoresistors by diffusion or ion implantation were constructed with a Wheatstone bridge. Then, the film was insulated electrically. The variation of piezoresistors caused by the applied pressure was transferred into voltage due to the piezoresistive effect. Single crystal silicon (Si), polysilicon (polySi) [7], graphene, and Si_3_N_4_ are frequently made into pressure sensitive diaphragms in piezoresistive sensors. Since the metal strain gauge was replaced by silicon piezoresistors, the size of the pressure sensors has been gradually reduced from the centimeter to the millimeter level.

For the capacitive pressure sensor, taking the parallel plate capacitor as an example, the distance between the two electrodes was varied by the deformation of the film electrode under pressure, so as to obtain the change of capacitance [8,9,10,11]. Figure 1b illustrates a thin film capacitive transducer with the metalized diaphragm structure over a conductive base [12]. The upper and lower surfaces with metal wires formed the two electrodes. According to the equation C = ε_0_ε_r_S/D [13], the capacitance was related to the distance D and the face-to-face area S between the two electrodes. As the pressure increased, the distance between the diaphragm and the substrate gradually decreased, and the value of the capacitor increased correspondingly. Common capacitive pressure sensors adopt Si, polySi, and SiO_2_ as sensitive diaphragm materials. By ion implantation or metal deposition, face-to-face electrodes can be obtained.

For the optical fiber pressure sensor, a Fabry-Perot (F-P) interferometer [14,15,16] was composed of a semi-reflector at the optical fiber end and a movable reflector (thin film). The structure of an optical fiber sensor [17] is shown in Figure 1c. The optical fiber and diaphragm were connected together through capillary tubing. Light was fired at a section of the fiber and reflected on a small interferometer at the end of the fiber and the inside surface of the diaphragm. The movable film would deform with applied pressure. The two reflected light would interfere through the beam splitter and aperture [18]. The deformation of the thin film caused by pressure changed the light path between the two mirrors, so the phase of the reflected light would be transformed. A photodetector was used to detect the reflected light intensity, which was associated with a displacement of the movable film. Silica glass, SiO_2_, and graphene [19] have been commonly used as sensing films for optical fiber pressure sensors. The diaphragm could be made with a size of dozens or hundreds of microns in diameter.

For the resonant pressure sensor, the pressure sensing element converted the applied pressure into the oscillation frequency change of the resonator [20,21,22]. As an important part of the structure, the resonator was fixed on a diaphragm or the edge of the cavity [23]. For example, an “H”-type resonant beam was suspended on the silicon pressure sensitive diaphragm with four fixing endpoints (Figure 1d). A magnetic field was used to excite the resonator, and the inductive voltage was used to detect the frequency of the resonator. The silicon pressure sensitive diaphragm deformed under the applied pressure, which caused tensile stress and compressive stress to act on the “H”-type resonator. Hence, the readout resonant frequency was related to the applied pressure. Silicon, polySi, quartz, and SiC could be used to fabricate resonant pressure sensors [24,25].

The piezoelectric pressure sensor was based on the piezoelectric effect. When the pressure was applied to the piezoelectric film, the bending film generated a potential difference on two surfaces [26,27]. Figure 1e shows a typical piezoelectric pressure sensor [28]. The sensor was composed of 0.1 μm thick platinum, 1 μm thick aluminum nitride (AlN), and 9 μm thick polyester (PET) film. AlN is a piezoelectric ceramic material. It is smooth and transparent. PET was used to separate the platinum electrodes to prevent accidental leakage of charge. Figure 1f shows the cross-section of the AlN layer. The AlN layer was composed of numerous oriented columnar fine crystals perpendicular to the surface of the mylar. The particle size was about 80 nm. With the help of a measuring circuit, the amplifying charge would be converted into the output of electron quantity, which was proportional to the applied pressure. Due to the excellent piezoelectric properties, piezoelectric ceramic materials, such as PZT, BiFeO_3_, ZnO, and AlN [29,30,31], have been often used as piezoelectric transducers. The film needed enough area to get a strong electrical signal because of the difficulty and complexity of charge collection. The area of piezoelectric pressure sensors is usually larger than 2 mm × 2 mm.

Miniaturization is one of the important parameters in the development of micro pressure sensors and has been given more attention. Miniature pressure sensors with a small size and low cost have been widely used in the medical, aerospace, industrial, automotive. and consumer electronics fields [32,33,34,35]. Commercial small pressure sensors have been developed, such as BMP380 from Bosch and LPS22HB from ST Microelectronics, total package sizes of the sensors being below 2 mm × 2 mm × 0.75 mm. From reported academic papers, the smallest pressure sensor was fabricated with graphene squeeze film, which was 5 μm in diameter and 10.5 nm in thickness. Based on resonant frequency shifting of graphene film with pressure, the sensor had a sensitivity of 9 kHz/mbar in the range of 8–1000 mbar. Due to the enormous demands of the above applications, miniature pressure sensors have attracted a large amount of research interest. However, the worse performance and difficult fabrication procedures are inevitable problems in the minimization of the pressure sensor. This paper reviews those latest achievements in the medical treatment, aerospace, industrial, automotive, and consumer electronics fields. Previously published reviews focused more on the application of miniature pressure sensors in a certain field [34,36,37] and rarely described the problems faced on the road to miniaturization and the possible future development direction in detail.

This paper includes a variety of working mechanisms for pressure sensors, including piezoresistive, capacitive, fiber optic, and resonant mechanisms. The miniaturization of pressure sensors is reviewed through the development of micromachining technology. This article focuses on the application of miniature pressure sensors in the medical field, as well as in the aerospace, industrial, aerospace, and consumer electronics fields. Then, the problems and corresponding solutions of pressure sensors in miniaturization are pointed out. Finally, the development direction of the miniature pressure sensor is given.

## 2. Size Change with Process Development

Silicon based pressure sensors and production of the cavity were taken as examples to illustrate the contribution of the following processes associated with the miniaturization. Since ion doping of silicon was found to have an excellent piezoresistive effect [26,38], the piezoresistive factor was 100 times higher than common metals, and the metal strain gauge was gradually replaced by silicon [39]. From then, the fabrication of silicon pressure sensors was optimized step by step.

In the 1950s, isotropic etching technology was developed to etch silicon wafers at the same rate in any orientation with HF, HNO_3_, and other corrosive liquids [40]. However, isotropic etching was hard to control and frequently led to nonnegligible etching error. Anisotropic wet etching technology could be traced back to the 1960s at Bell Lab. A mixed solution of KOH, ethanol, and water [41] for anisotropic etching of silicon was used. Then, many kinds of organic and inorganic etchants were developed in succession [26]. Anisotropic etching fluids have different corrosion rates on different crystal surfaces. Based on this corrosion characteristic, a variety of microstructures could be fabricated on silicon substrates.

The crystal orientation and concentration dependence of the silicon etching rate about the (100) and (111) crystal planes of silicon, as well as chemically anisotropic etching were discussed [42]. For KOH etching solution, the results of different etching processes of silicon are presented in Figure 2. It had the highest etching rate for the (100) crystal plane, followed by the (110) crystal plane, and the lowest etching rate for (111) crystal plane. Figure 2a shows that if a silicon substrate with the (100) crystal plane was etched, a trapezoidal groove with an inclination angle of 54.74 degrees was formed. For both (110) and (111) crystal surfaces, the etching rate was relatively slow, so the corrosion progress of the sidewall and bottom was similar (Figure 2b). If the etching window was small, the sidewall was not corroded sufficiently, and a wedge groove would be formed. For an isotropic etching solution, it had similar etching rates in all directions, forming a hemispherical or elliptical groove (Figure 2c).

The first three-dimensional micromechanical silicon piezoresistive pressure sensor were made by bonding technology, which allowed the film’s periphery to be anchored to a medical catheter without stress concentration [43,44]. The diaphragm was formed by isotropic corrosion of a silicon wafer. The thickness of the diaphragm was determined by electrochemical etching. The thick walled glass tube supported the diaphragm through its clamping ring, thereby allowing the steady placement of the thin diaphragm. The diameter of the diaphragm ranged from 0.3–2 mm, and the thickness of the diaphragm ranged from 5–50 μm.

In the 1970s, the further development of bonding technology promoted the continuous reduction of chip size. Different from the defects of large stress and easy creep caused by adhesion, bonding technology had the advantages of good tightness, small profit, high bonding strength, and a simple process [45]. Anodic bonding technology was used to bond thin films with glass support together under vacuum and produced the first absolute pressure capacitor transducer [46], which had better thermal stability [47]. A parallel plate capacitor was formed by the upper electrode (the film) and the lower electrode (the bottom of a cavity). Although the square diaphragm was 25 μm thick, the electrode gap was 2 μm, and the length was 1.5 mm, the usage of anodic bonding technology to airtightly seal the reference pressure chamber would eliminate the need for a large reference pressure port (such as a catheter). Therefore, the overall size of the sensing device decreased obviously, while the thickness was twice as much as before. However, based on the reliability of bonding technology and chemical dissolution technology, the thickness could be further reduced. In 1991, a circuit compensation system [48] was combined with the capacitive pressure sensors with the elimination of the thick supporting edge around a thin film to produce a chip area of 1.1 mm × 0.45 mm. By utilizing chemical mechanical polishing (CMP) [49,50,51], the device was further thinned to 262 μm.

A manufacturing process of a touch capacitive pressure sensor was adopted to shorten the assembly mode of the two components of substrates and films [52]. Some bonding techniques for pressure sensors with different substrate materials are generalized in Table 1. Figure 3 shows a silicon-glass capacitive pressure sensor using the anodic bonding technique to assemble the silicon diaphragm and the glass substrate. When pressure was loaded on the film, the distance of the gap changed. Then, the vibration of capacitance could be reflected through the upper and lower electrodes. During this period, both capacitive and piezoresistive pressure sensors had similar structures depending on the bonding technology. The difference was that the capacitive pressure sensor had more electrodes at the bottom of the cavity than the latter.

The above processes such as isotropic and anisotropic corrosion, adhesion, and bonding, all belonged to bulk silicon processing technology. In the 1980s and 1990s, surface microfabrication processes had resulted in the rapid development of MEMS technologies [53,54]. The process started with the deposition of a thin sacrificial structure layer on the substrate. Then, a structural layer was deposited on the sacrificial layer. The circuit was then laid on the substrate, and the sacrificial layer was etched away, creating a cavity to form a three-dimensional structure [55]. The thickness of the film could be controlled accurately, and the surface microfabrication process might have eliminated adhesion or bonding steps, which led to a further reduction in sensor size.

One of representative pressure sensor was reported in [56], which was manufactured with the following process, as shown in Figure 4: (I) deposited polySi on silicon substrate as the sacrificial layer, (II) deposited Si_3_N_4_ on the polySi layer as the pressure sensitive diaphragm, and (III) deposited another polySi on Si_3_N_4_ as the piezoresistive layer. (IV) The diaphragm of the pressure sensor with the cavity was formed by releasing the sacrificial layer through etch-holes. The diameter of the pressure sensitive diaphragm was 100 μm, and the thickness was 1.6 μm. Under the pressure range of 0–300 kPa, the pressure sensitivity of 10 μV/V/kPa was obtained. For adopting Si_3_N_4_ films similarly, a piezoresistive pressure sensor with a flat surface was produced by combining bulk silicon with surface processing technology [57,58]. A cavity was made in the surface of the silicon, and SiO_2_ was deposited to fill it, which was flattened by CMP later. Then, a planar sensor with sacrificial oxide grooves was achieved. A diaphragm of 100 µm in diameter, but a thickness of 0.8 μm, is observed in Figure 5a [59]. Six radial and one circumferential piezoresistors were formed by implanting phosphorus. Absolute pressure was detected with the reference pressure chamber below the diaphragm (Figure 5b). When pressure was applied, the diaphragm deformed, resulting in an equal, but opposite change in sign between the radial and circumferential strain glue. For the full-bridge configuration, a pressure sensitivity of 120–140 μV/V/psi would be obtained.

One electrode of a capacitive pressure sensor on the substrate was formed [60] by selective ion implantation, and then, silica was deposited as a sacrificial layer. The PolySi layer was deposited on the sacrificial layer as the upper electrode of the capacitor. The capacitive pressure sensor was formed by etching the sacrificial layer through corrosion channels. The surface morphology of the sensor after surface micromechanical processing is shown in Figure 5c. The circular diaphragms ranged from 50 to 120 μm in diameter, and the thickness was 1.5 μm. The gap between the two electrodes was 900 nm, and the enlarged view of it is shown in Figure 5d. Due to the weak signal from capacitance change, the accompanying integrated circuit (IC) system was necessary, which resulted in the final device size of 1.1 mm × 0.7 mm.

A modular manufacturing method of polySi surface micromachining structure was proposed in [61]. This technique included tungsten metallization, low pressure chemical vapor deposition (LPCVD) oxide and nitride passivation, planarization of sacrificial spin-on-glasses, and final release of the structure in hydrofluoric acid. In Figure 5e, a scanning electron microscope (SEM) image of the released polySi MEMS structure is demonstrated [62]. The device consisted of three suspended and movable layers. The Sacrificial polysilicon (SP1) layer represented circuit-to-structure connectors. SP2 was interdigitated comb drives, suspension springs, fuses, and the mechanical plate. SP3 was the upper electrode. SG represents the sacrificial glass. At some locations, a certain thickness of sacrificial phosphosilicate glass (PSG) was deposited to form stand-offs (SD) with subsequent structural polySi. These SDs were used as a limit stop to prevent big deformation of the structural layers. The HF was allowed to pass through the etched holes into the sacrificial SG below the structural layer. Tungsten passivation metallization provided convenience for multilayer structures during final release etching.

To make a hollow cavity in the pressure sensor, instead of the above process, the Bosch laboratory developed an APSM process (advanced porous silicon membrane) based on the reorganization of porous silicon for pressure sensors [64,65,66,67]. The process is shown in Figure 6a [36]. (I) The electrochemical corrosion of the defined p-region silicon was carried out to form a layer of nanoporous silicon. (II) The annealing at above 400 °C for a long time could rearrange the porous silicon layer, and a vacuum cavity was formed. (III) Epitaxial growth was used to seal the surface in high vacuum. The buried vacuum cavity had a depth of 4 μm. The epitaxial membrane had a thickness of 9 μm and a lateral dimension of 550 μm × 550 μm. ST Microelectronics laboratory [68] proposed selective epitaxial growth in monocrystalline silicon to form buried channels, which also could be used to fabricate cavities. The production process was as follows (Figure 6b). (I) After initial lithography, the silicon wafer was selectively oxidized. Then, the wafer was introduced into the epitaxy reactor for epitaxial growth. (II) The epitaxial growth of single crystals began at the crystal axis of the silicon lattice and diffused over the oxide layer. (III) After annealing under hydrogen atmosphere at a high temperature of 1080 °C, SiO was formed at the Si-SiO_2_ interface. (IV) In fact, SiO was a very volatile element that produced a void at the Si-SiO_2_ interface. Therefore, after a long time of annealing, a buried channel was formed. LOCOS (local oxidation of silicon) isolation was used to protect the epitaxial growth in monocrystalline silicon, followed by piezoresistors and interconnect implantation. Then, a piezoresistive pressure sensor with the silicon on nothing (SON) structure would be produced. Neither process required the wafer-to-wafer bonding process to seal the cavity. Due to the surface migration resulting from minimizing surface energy, the groove could be transformed into the empty space in Si. Under the condition of controlling the aperture and annealing temperature, the SON structure could be made [69,70].

With unceasing development of microfabrication technologies, in the 1980s and 1990s, deep reactive ion etching (DRIE) and silicon on insulator (SOI) emerged and were used to manufacture pressure sensors. For example, DRIE could be used to make grooves with high depth-to-width ratios (300 μm deep) and high steepness, which played an important role in reducing the opening area of the cavity [71]. Using DRIE, mass production of complicated structures was realized. For example, Figure 7a shows the structure of an electrostatic resonator. The multi-folded structure was the resonant spring of the resonator. The separation between the resonant spring and the cavity s clearly visible. Figure 7b shows an SEM image of a thermal actuator. The device was operated by an expansion of two heating resistors (to the right of this picture) that were pushed from opposite directions relative to a third beam at a slightly offset point. Combined with silicon fusion bonding, more complex three-dimensional structures could be produced. Doping ions were implanted into the device layer of SOI to form piezoresistors [72]. Then, the silicon handler of SOI was thinned to 150 μm, and a cavity with the vertical wall was etched by DRIE. The sensor had a diaphragm with a diameter of 92 μm and thickness of 2.5 μm and a total size of 680 μm × 260 μm × 150 μm. Configured on the Wheatstone bridge, the sensitivity of sensors was determined to be 27–31 mV/V/mmHg.

A MEMS manufacturing process of absolute pressure resonant sensors was introduced in [73]. DRIE technology was used to fabricate the microstructure; meanwhile, the etching step and passivation step in the etching process were optimized to obtain the high depth-to-width ratio. Finally, due to high selectivity with 1/150 for conventional resist and 1/200 for SiO_2_, 2 μm critical dimensions were achieved through etching over a 20 μm thick SOI layer. The profiles of different capacitive resonator structure were observed. Figure 7c shows straight beams with a section of 20 μm × 30 μm. Figure 7d shows comb beams with 2 μm critical dimensions. The change of pressure resulted in the variation of the relative position between these beams and the device. Different from using the device layer of SOI as the pressure sensitive film, polySi films and piezoresistors were also fabricated. Through the deposition of polySi film on SOI, a piezoresistive pressure sensor was made [74]. Figure 7e demonstrates another piezoresistive film, a 1.2 μm thick polySi layer. The sensitive film had an outer radius of 80 μm. Piezoresistors were formed by ion implantation and etching the device layer. Through HF etching, a cavity was formed by eliminating the SOI buried oxide layer. PolySi films and thick dielectric layers were used as center plugs and metalized protection, reducing the risk of cracks in the sensor structure.

Nanowires, such as Si/Au/Ag/SiC [75] nanowires, could be etched out by RIE or grown by MBE (molecular beam epitaxy). These nanowires had mechanical flexibility and could be used to fabricate new sensitive elements in pressure sensors [76,77]. A thin film pressure sensor with embedded silicon nanowire (SiNW) piezoresistors [78,79] was produced. The ultra-sensitive SiNWs were formed (Figure 7f) with a size of 10 μm × 100 nm × 100 nm by etching the device layer of p-type (100) SOI. After opening a window for depositing polySi on the device layer, SiO_2_ was deposited to form a membrane and fill the trenches, which defined the edge of the cavity. The cavity was eventually released by etching silicon and the polySi on the back. From Figure 7g, the four pairs of SiNWs were located in four symmetrical radial positions on the circular membrane.

Whether Si based or a non-silicon pressure sensor, the processing technology played a decisive role in reducing the size of the pressure sensors. The production and improvement of semiconductor processing technology not only accelerated the development of new pressure sensors, but also promoted the gradual reduction of device size. The size of pressure sensors had also been reduced from the centimeter level in the middle of the last century to the micron level currently. The length of the pressure sensitive diaphragm reduced to less than 100 μm from over 1 cm. In particular, the diameter of the pressure sensitive diaphragm, made of graphene, could be less than 30 μm, even down to 5 μm.

## 3. Application of Small Pressure Sensors

As mentioned above, the size of pressure sensors was limited by the requirements of the application environments, which pushed the development of pressure sensors to a smaller size. In the following, small pressure sensors used in the medical, aerospace, industrial, and consumer electronics fields are discussed. The device material, structure, fabricating processes, and performance of each sensor will be illustrated.

### 3.1. Medical Applications

The classic application of miniature pressure sensors used in the medical field was the measurement of carotid blood pressure of horses [80,81]. With the development of micro-machining technology, thin films with high length: thickness ratio were made. After the improvement of the processing techniques described in the previous section, a variety of small pressure sensors could be used in the medical field. This section will illustrate the application of small sized pressure sensors in the medical field with common applications such as blood pressure, intraocular pressure, intracranial pressure, and the biocompatibility issue.

#### 3.1.1. Blood Pressure

Currently, most common vascular diseases are caused by vascular blockages, such as coronary heart disease. In modern medicine, arteriography is a routine means for medical workers to diagnose coronary heart disease and hypertension. Therefore, miniature pressure sensors could be used to implant medical catheters into vessels with suspected lesions for intraoperative or long term pressure monitoring [82,83,84,85]. The thin blood vessels and complex stenotic lesions put high demands on the size of pressure sensors. In particular, the pressure sensors were limited in both length and thickness due to the characteristics of long and thin blood vessels.

An ultra-small capacitive pressure sensor with 3 μm electrode spacing was produced [86]. The 12 μm thick edge region of the sensor was determined by deep boron doping. Then, a shallower boron diffusion was used to define the 1.5 μm thick diaphragm. A dielectric was then deposited and patterned to provide subsequent protection. The silicon wafer was bonded to a Corning 7740 borosilicate glass plate with a patterned metal region. Finally, the device was immersed in an anisotropic etching agent of silicon, where partial silicon was dissolved, and only the boron doped part of the wafer was retained. The size of the diaphragm was 290 μm × 550 μm × 1.5 μm, and the sensitivity was 1.39 fF/mmHg at 500 mmHg. The whole device could be installed in a 0.5 mm outer diameter catheter, which was suitable for measuring multi-point pressure from the coronary artery of the heart. Compared with the above work, the chip size was further reduced in [87], and the critical width and thickness of the sensor were reduced from 500 μm to 350 μm and 250 μm to 100 μm, respectively. In contrast with being sealed at atmospheric pressure, the device sealed in a vacuum avoided gas damping and thermal expansion, improving its pressure sensitivity. The small capacitive pressure sensor had a size of 1.4 mm × 350 μm × 100 μm. The length, width, and thickness of the diaphragm were 420 μm × 220 μm × 1.6 μm, and the largest pressure sensitivity reached 3 fF/mmHg in the pressure range of 0–1000 mmHg.

A piezoresistive pressure sensor with a vacuum cavity was made by releasing sacrificial silica. The cavity and channels were made on the substrate by silicon etching and SiO_2_ thermally growing to produce a flat surface [88]. SiO_2_ and polySi were deposited as the sensitive film. Then, depositing the dielectric layer and the second layer of polySi, piezoresistors were produced by ion implantation. Finally, the cavity could be produced by immersing in HF solution. The film had a thickness of 400 nm and a width of 103 μm. The size of the whole device after dicing was 1300 μm × 130 μm × 100 μm. At the range of −25 mmHg to 300 mmHg, the sensor sensitivity was /2.0 μV/V/mmHg.

An absolute piezoresistive pressure sensor was made by anodic bonding [89]. Piezoresistors were formed on the surface of silicon wafer after selective ion implantation. The total structure comprised lead wires, a sensing surface with a piezoresistive layer, and a top glass bonded to the chip to provide a reference pressure. A side sketch of the structure is schematically shown in Figure 8a. The size of the diaphragm was 280 μm × 130 μm × 5 μm. The device’s overall size was 1.1 mm × 240 μm × 74 μm with a half-bridge structure. The actual cross-section and the sensor die are shown in Figure 8b, with the side-view of the diced sensor showing the glass cap on the left and thin silicon substrate on the right. The production process was as follows: (I) Silicon dioxide was deposited onto a silicon wafer. (II) Two ion implantations were performed to produce piezoresistors and a high doping region. (III) Etching a v-groove out and making a welding pad in the v-groove area with metallization were performed. (IV) Etching silicon to form the cavity by back optical alignment was performed. (VI) Glass substrate with grooves and cavities bonded to the wafer by anodic bonding. After CMP, a thickness of 74 μm could be achieved. The pressure sensor was manufactured in the guidewire (a diameter of 0.35 mm) to allow accurate measurement of the pressure in the coronary artery.

A temperature compensated dual beam pressure sensor was presented. The cavity was etched out on a silicon wafer with KOH. Next, the size of the vacuum chamber was defined by the filling of LOCOS silicon oxide. The channels were formed by the deposition and definition of silicon dioxide. Low stress Si_3_N_4_ and polySi were deposited and patterned to form the two beams. On the beams, a polySi piezoresistor was produced above an insulating low stress Si_3_N_4_ layer. Then, another SiO_2_ layer of a 1.6 μm thickness was deposited to define the spacing between the diaphragm and the beam. The SiO_2_ was also used as the beam to diaphragm attachment. Then, low stress Si_3_N_4_ and polySi were deposited to serve as the diaphragm [90,91]. SiO_2_ was etched by immersing in a 50% HF solution, to release the beams and diaphragm. Finally, after sealing at low pressure in an LPCVD process step, the sealed beam-type resonant pressure sensor was obtained. Beams were completely enclosed in the reference vacuum cavity formed under the diaphragm. Figure 9a gave a schematic drawing of the piezoresistive sensor. A piezoresistor and a reference resistor of the same resistance value were placed on the pressure sensitive and temperature compensation beam, respectively. These beams were connected to the film by means of attachments. The piezoresistors were placed on the 100 μm long cross-beam to achieve thermal matching. Different from pressure sensors in which reference resistors were placed on the substrate, the pressure sensor with temperature sensing piezoresistors on the beam possessed accurate temperature matching [92]. The diaphragm had a side length of 100 μm with a thickness of 2 μm, and the beams had a width of 40 μm and a thickness of 1 μm (Figure 9b). The sensitivity of the ultra-small pressure sensor was 0.8 μV/V/mmHg.

In addition to piezoresistive and capacitive sensors, optical fiber pressure sensors were also of great concern due to their advantages of anti-electromagnetic disturbance, being lightweight, and having high sensitivity [93,94,95]. An optical fiber pressure sensor was developed for medical pressure measurement [96,97]. The mesa with 2.3 μm thickness and 60 μm in diameter was obtained by depositing SiO_2_ on a silicon wafer and then patterned. The aluminum was deposited on the mesa serving as a mirror. The adhesive spacer was formed with photosensitive polyimide [98]. In the pressure range from −300 mmHg to +300 mmHg, the maximum variation of reflected light intensity arrived at 64%. The periodical vibrations with increasing pressure were consistent with general optical interference results, indicating that the interferometer (sensing element) in optical fiber had the function of a reflecting pressure sensor.

An optical fiber tip F-P interferometric pressure sensor based on an in situ μ-printed air cavity was presented [99]. A suspended SU-8 diaphragm [100] was fabricated on the end face of a standard single mode optical fiber to form a miniature F-P cavity with an in-house optical 3D μ-printing setup. The fiber was mounted on a ceramic ferrule, and the outer and inner support walls had a radius of 100 μm and 80 μm, respectively. When the pressure was applied to the diaphragm, the length of the F-P cavity changed with the deflection. Microcolumn arrays were printed on the diaphragm as light scatterers to suppress the light reflection at the outer surface of the polymer cavity. The optical fiber was mounted in a ceramic ferrule with the inner radius of 80 μm. In the range of 0–700 kPa, a sensitivity of 2.93 nm/MPa was obtained. A miniature fiber pressure sensor based on an in-fiber confocal cavity was proposed [101]. The fabrication process of the cavity and thin film also included fusion splicing, cleaving, and polishing. The micro-confocal air cavity [102] was formed by splicing a single mode fiber (SMF) and a solid-core photonic crystal fiber together for high contrast and high resolution. By tracing the trough wavelength in the interference spectrum, the measurement was performed, and under the pressure range of 0–1 MPa, a sensitivity of 193 pm/MPa was obtained.

#### 3.1.2. Intraocular Pressure

Many eye diseases around the world are related to high intraocular pressure (IOP) [103], which means that the measurement of intraocular pressure is fully essential for the diagnosis and treatment of these diseases. Different from slender blood vessels, the size of pressure sensors was strictly limited by the measurement of intraocular pressure because of the curved forward structure of the eyes and the extremely sensitive eye tissues [104]. However, compared with the blood pressure sensor, the size of the sensor used for intraocular pressure measurement was slightly larger than 2 mm × 2 mm [105,106]. Taking into account the transmission of pressure data, the electrical signal from the pressure sensors used for intraocular pressure measurement was usually transmitted wirelessly.

Si_3_N_4_, phospho-silicate glass (PSG), and the doped polySi were deposited on a silicon wafer as the pressure sensitive film, the sacrificial layer, and electrode, respectively. By HF etching PSG, a capacitive pressure sensor was produced [107], and the film size was 100 μm × 100 μm. A series LC resonator was constructed with a capacitor array of 32 membranes and a six turn hand rounded AWG38 enamel coil [108]. By using a middle LC resonator, a sensitivity of 18.5 Hz/mmHg could be obtained at the detection distance of 1.5 cm. A film bulk acoustic resonator pressure sensor was proposed with ZnO piezoelectric film [109]. The sequential deposition of silica/Cr/Au/ZnO films on a silicon substrate formed a membrane and electrodes with the help of an innovative high target utilization sputtering (HiTUS) system [110,111]. By etching silicon on the back, an acoustic resonator could be made with a cavity. The SiO_2_/ZnO membrane had the dimensions of 200 μm × 200 μm. Within the range of 0–40 kPa, the pressure sensitivity was 6.1 ppm/kPa.

A MEMS capacitive pressure sensor based on a p++ silicon membrane was designed and characterized. According to the different conditions of fixed support around the diaphragm, structures were divided into the clamped diaphragm and grooved diaphragm. Both diaphragms had a size of 0.55 mm × 0.55 mm and a thickness of 4 μm. In this design, there was a cavity with a depth of 1.5 μm between the diaphragm and the Pyrex glass backplate with a gold backplate electrode [112]. Through the finite element simulation, it was found that the suspension of the groove diaphragm along its periphery had a great influence on the mechanical sensitivity. The suspended structure was achieved by cutting grooves in the diaphragm, so that the diaphragm was supported only on a short suspension. This structure would reduce residual stress and diaphragm stiffness [113]. In the range of 0–60 mmHg (the pressure range of common intraocular pressure sensors), the sensitivity with capacitive change was 18.11 ppm/Pa for the clamped pressure and 59.6 ppm/Pa for the slotted pressure.

Conventional sensors were subject to rigidity during implantation into the human body, resulting in size limitations in the narrow lesion area. In contrast, flexible sensors could be placed more accurately in the viewing zone, producing more efficient pressure signals. Polylactic acid (PLA), parylene, and liquid crystal polymer were the common polymer biomaterials [114]. A capacitive pressure sensor based on ultrathin flexible polymer was fabricated. The substrate was a soft and flexible liquid crystal polymer (LCP) with a 25 μm thickness [115]. Using a sacrificial photoresist (PR) layer, the substrate was bonded to a carrier wafer. The “sandwich” structure (parylene layer-Ti/Au-parylene layer) included a pressure sensitive film on the top of LCP. Ti/Au served as the upper electrode. The film was almost flat with a surface roughness of 0.2–0.3 μm. By immersing in acetone, the substrate and the carrier wafer were separated. Then, DRIE was used to etch the backside LCP. Finally, a cavity was produced by using 50 μm thick Kapton to LCP backside sealing. The circular cavity had a diameter of 260 microns. Figure 10a shows the surface profile of the capacitive pressure sensor fabricated by the membrane transfer technique. The sensing area was about 300 μm × 300 μm, and with the integrated circuit, the sensor parts volume was 700 μm × 700 μm × 150 μm. The device was placed in the eyes of mice to measure intraocular pressure, and the results showed a sensitivity of 3.3 fF/mmHg, which was comparable to commercial sensors with a large diaphragm.

A small implantable optical pressure sensor that could provide convenient, accurate, on-demand IOP monitoring in the home environment was reported. From top to bottom, the sensor was composed of a flexible Si_3_N_4_ membrane, nanodot array, membrane carrier, and bottom silicon mirror (Figure 10b). When the IOP increased or decreased, the membrane deflected inward or outward, leading to a new resonance spectral signal, which was obtained by using a commercial miniature spectrometer for signal analysis. It worked by taking advantage of its own gap and reflecting a characteristic resonance spectral feature under the intraocular pressure with the bottom silicon mirror. Owing to the high optical transparency, large refractive index, and robust mechanical resilience, the Si_3_N_4_ membrane was chosen as the membrane for sensing pressure. The gold nanodot array was used to optimize the reflectivity at the top of the thin film to match the reflectivity at the bottom of the silicon surface. Figure 9c shows the cross-sectional schematic view of the sensor, and the inset is the image of the nanodot array. The diameter of each dot was 600 nm. The oversized sensor was 900 μm in diameter and 600 μm in thickness. The sensor with the nanodots showed an average accuracy of 0.29 mmHg over the range of 0–40 mmHg at a 3–5 cm readout distance [2].

The data acquisition systems for data transmission in IOP sensors were also developed. An active IOP monitoring system was developed that collected pressure data through MEMS sensors and an application specific integrated circuit (ASIC) chip [116]. The data could be transmitted directly to external devices or stored in on-chip memory. Another IOP telemetry system was developed for continuous wireless intraocular pressure and body temperature measurement for small animals. This system was composed of a coupler, connection tube, piezoresistive pressure sensor, temperature sensor, amplifier, microcontroller, wireless transmitter, and power circuit. The system could record IOP with 0.3 mmHg accuracy and negligible drift at a rate of 0.25 Hz for 1–2 months [117].

#### 3.1.3. Intracranial Pressure

Intracranial pressure (ICP) monitoring is essential in the diagnosis of inflammatory diseases of the central nervous system. In addition to the common CT angiography, lumbar puncture is a method to measure intracranial pressure by lumbar intervertebral puncture and removing cerebrospinal fluid for examination. It is also a minimally invasive method for clinical ICP monitoring [118,119,120]. The traditional lumbar puncture was measured by draining cerebrospinal fluid to an external u-shaped tube through a puncture needle. This method was complex, time consuming, and may have caused secondary trauma and injury to the patient. With the development of micromachining technology, small pressure sensors could be installed [121] into the puncture needle for pressure measurement, which makes the ICP measurement convenient and accurate and reduces the pain of patients.

A small piezoresistive pressure sensor was fabricated with piezoresistors by ion implantation, etched cavities, and anodic bonding on the back [122]. The pressure sensitive film had a 2 μm thickness and was no longer than 100 μm. The total device area was no more than 300 μm × 300 μm. Test results showed that within the 0–25 kPa range, which contained the intracranial pressure range of a supine adult (0–2 kPa), the sensitivity was 2.85 μV/kPa and the nonlinearity was 4.67% FS.

Laser cutting or HF wet chemical etching could prepare a thin film [123]. The SiO_2_ membrane was connected to the end of the glass tube through ultraviolet (UV) adhesive. The polished optical fiber was placed in the glass tube and aligned with the center of the diaphragm. The deformation of the diaphragm under the applied pressure resulted in interference patterns between two surfaces of the cavity. Based on the principle of multi-beam interference and the F-P interferometer, the relationship between reflected light intensity and the pressure was established. The sensor was encapsulated in a probe with a long glass tube. The sensitivity was 39.2 nm/kPa and 99.5% with a resolution of 0.13 kPa, which could meet the requirements of the ICP measurement in the range of 0–25 kPa.

#### 3.1.4. Biocompatibility Issue

Blood pressure/intraocular pressure/intracranial pressure measurements were mostly short term or intraoperative real-time measurements. For long term implantable pressure sensors, the influence of biological environment should be considered [124,125,126]. Therefore, except the premise of small size, the protection of sensitive units should be contained in the manufacturing process. One approach was to add a biocompatible coating on the surface of the sensor, which could not only solve the problem of biological rejection, but also protect the bare piezoresistive resistance from leakage. An absolute capacitive pressure sensor with a 400 nm thick and 70 μm^2^ polySi diaphragm was made by surface micromechanical processing technology [127]. The miniaturization and integration by integrating the surface micromechanical pressure sensor into the advanced technology of the standard CMOS/BiCMOS (Bipolar junction transistor) process was achieved [128]. For reasons of protection and biocompatibility, the sensor was coated with a silicone elastomer with a thickness of 100 μm, which did not affect the performance of the sensor. By combining the sensor with CMOS technology, a new low power integrated pressure sensor system with digital output signal was obtained. The entire system was less than 500 μm in thickness and 0.8 mm × 3.8 mm in width and length. The output sensitivity of 5.9 digits/mmHg was shown under the pressure of 500–1100 mmHg.

Another way to protect the resistors was to place the piezoresistors under the diaphragm, as many sensors had done before. This built-in isolation prevented electrical resistance from contacting the biological environment. For instance, the SOI device layer was bonded to an etched glass substrate after ion implantation to make the piezoresistors at the interface of the diaphragm and the cavity, so that the piezoresistors would not touch the biological environment [129]. The pressure sensors were built to 820 μm × 820 μm × 500 μm. The sensitivity was 20 mV/V/mbar in the range of 500–1500 mbar.

In contrast, bioabsorbable pressure sensors could reduce secondary injury to patients. Bioresorbable silicon pressure sensors for the brain were reported [130]. All components of these sensors were naturally absorbed by hydrolysis/metabolism without extraction [131,132,133]. Figure 11a shows the structure, which involved a polylactic acid-glycolic acid (PLGA, 30 μm thick) film, a silicon nanomembrane, and a cavity (depth of 30–40 μm) sealed in nanoporous silicon (60–80 μm thick; 71% porous). The cavity was sealed between PLGA film and the substrate of nanoporous silicon. Silicon nanofilm with a serpentine structure acted as a piezoresistive element and was located at one edge of the film. The fabrication process was as follows: Nanoporous silicon with about an 80 μm thickness was prepared by double polishing the p-type silicon wafer at a current density of 160 mA/cm^2^. Then, nanoporous silicon was transferred to PDMS, and a cavity was subsequently etched. Si-NM was produced by solid state diffusion of boron into the SOI wafer. By etching the buried oxygen layer with HF, it was transferred to a bilayer of diluted polyimide/PMMA as a temporary silicon carrier substrate. Then, SiO_2_ with 100 nm was deposited as a passivation layer, and all layers were transferred to a film of PLGA (lactide:glycolide composition with 3:1 wt %). Finally, the air cavity was sealed by heating and laminating the PLGA to nanoporous silicon. Figure 11b shows the schematic illustration of the sensor. The substrate had a square cavity with a depth of 30–40 μm, and the inset shows the location of the silicon nanomembrane (Si-NMs). The silicon nanofilm as a piezoresistive element was etched into the snake-like geometry. Resistance varied linearly with pressure and an inclination of 83 Ω/mm Hg (Figure 11c). The entire device is shown in Figure 11d, with a total size of 2 mm × 1 mm × 0.08 mm (cavity size: 0.8 mm × 0.67mm × 0.03 mm) and the weight of 0.4 mg. When the device was inserted into the artificial cerebrospinal fluid, the silicon nanofilm, SiO_2_ layer, and nanoporous silicon would hydrolyze. After 15 hours, the silicon nanofilm and SiO_2_ layer first began to dissolve and disappeared after 30 hours, while the PLGA dissolved in four to five weeks.

### 3.2. Aerospace

In military and civil aviation applications, there are many places where miniature pressure sensors could be found, including flight data systems, environmental and tank pressures, hydraulic systems in the fuselage, and other applications such as hatches, oxygen hoods, flight tests, and structural monitoring [134]. The small pressure sensor with mass reduction and key point detection capabilities for spacecraft brought the benefits of reduced operating costs. From these perspectives, miniature pressure sensors could dominate the military and aerospace sectors.

A capacitive pressure sensor for spacecraft altimeter was designed [135]. Gold was selected as a pressure sensitive film to analyze center deflection [136]. The diaphragm had a size of 1000 μm × 1000 μm × 300 μm with a 3 μm thickness. Figure 12a shows the cross-section of the capacitive pressure sensor. Silicon was used as a binding substrate on which a 1 μm thick film of gold was deposited by the chemical vapor deposition (CVD) process to form an electrode. The silicon sidewall was used to separate the top and bottom electrodes of the capacitance sensor. When the pressure ranged from 100 mbar to 1100 mbar, the membrane deformation was linear. At low altitudes, capacitance sensitivity was 7.95 fF/mbar in the −50 to 30 °C temperature range.

With respect to pressure measurement of gas turbines, a high temperature resulted in inevitable failure to the sensors. In addition, silicon pressure sensors could not survive the corrosion and oxidation environment of jet engines and must be kept away from hot gas flow [137]. The doped SiC piezoresistive layer showed stable behavior at a high temperature (up to 800 °C) [138,139]. A SiC MEMS resonant strain sensor was developed with a silicon surface micromachining process for harsh environment applications [140,141]. SiC was deposited and patterned to produce a two terminal tuning fork structure. The size of the whole resonant structure was not more than 300 μm × 300 μm. A new SiC based touch pressure sensor was introduced [142]. The cavity was etched out on the glass substrate, and W was deposited as the lower electrode. SiC, W, and SiO_2_ layers were deposited in order on a silicon wafer, followed by selective etching to be the lower electrode. Glass was bonded to the silicon wafer by anodic bonding. When the width of the diaphragm was 200 μm, the sensitivity was 0.067 pF/kPa in the pressure range of 0.2–3.5 atm.

A piezoresistive absolute pressure sensor was designed and fabricated [143]. Piezoresistors were formed by ion implantation in the top layer of SOI. The cavity was fabricated through etching bottom silicon on the back. Finally, piezoresistive absolute pressure sensors were produced by bonding with the anode of glass. Pressure sensors with the applicable range of 0–400 bar were made by changing the thickness of the device layer. A photograph of the diced sensor with bonding wire is shown in Figure 12b. The front side with four piezoresistors and the backside with a cavity are shown, respectively. The size of the diaphragm was 750 μm × 750 μm × 210 μm, and the overall size of the device was 2 mm × 2 mm. The sensitivity was 0.35 mV/V/bar. For another piezoresistive SOI pressure sensor [144], a 15 μm groove was etched to define the thickness of the film, and Al was metalized on the upper layer. The film size was 500 μm × 500 μm × 15 μm. A pressure sensitivity of 3 mV/bar was obtained in the range of 0–5 bar by pressure measurement in the environment exposed to a 100 krads radiation dose. The zero drift was 0.08% full scale output (FSO) and not sensitive to temperature within 20–70 °C. PolySi based piezoresistive pressure sensors were also reported. With polySi deposition and ion implantation, the pressure sensors were completed by releasing sacrificial silica, and the final chip size was 800 μm × 800 μm. In the range of 0–100 psi, a sensitivity of 0.15 mV/V/Psi was obtained [145]. In the 40 °C to 120 °C temperature range, suitable for aerospace applications, the linearity was still good.

### 3.3. Industry

Due to the small size, low price, easy integration, and high reliability, miniature pressure sensors could be very suitable for the requirements of complex environments in industry [146,147]. Taking the automobile industry as an example, miniature pressure sensors would be applied for measuring the changes in the pressure of the auto brake system, the pressure of the tires, the fluid pressure of the transmission system, the pressure of engine oil, and the pressure of intake pipes.

The optical fiber pressure sensor was free from electromagnetic and RF interference and had the advantages of high temperature resistance and corrosion resistance, suitable for pressure testing in harsh environments [148,149,150]. The construction and test of spherical particles based on the orientation independent reflectivity were presented [151]. Each particle consisted of microfabricated spherical shells or micro-balloons. Figure 13a shows the cross-section of one micro-balloon and an amplification of the parylene shell with Al_2_O_3_ diffusion barrier. The diameter of each micro-balloon was a function of the pressure difference between the shell exterior and interior pressure. The thickness of 0.4 μm and diameter of 12 μm hollow flexible parylene shells with and without an ultrathin Al_2_O_3_ coating as diffusion barriers constituted the particles. The change of the particle radius was measured from particle spectral reflectivity. The radial pressure sensitivities of the micro balloons were 0.64 nm/psi and 0.44 nm/psi, respectively. Unlike the above mentioned spherical thin film sensing optical reflection, a flat thin diaphragm was also used as a movable reflective layer in an F-P interferometer. An ultra-high sensitivity gas pressure sensor based on the F-P interferometer with a fiber tip diaphragm was demonstrated. The structure was comprised of a silica capillary and ultrathin silica diaphragm with a thickness of 170 nm fabricated by an electrical arc discharge technique [152]. An SEM image of the ultrathin silica diaphragm is shown (Figure 13b), fused at the end of the silica capillary, having an inner diameter of 75 μm and an outer diameter of 125 μm. The pressure sensitivity was about 12.22 nm/kPa. Moreover, the pressure sensor showed a cross-sensitivity of about 106 Pa/°C at low temperature and functioned well up to a temperature of about 1000 °C, indicating that it could potentially be employed in a high temperature environment.

A fiber optic in-line Mach–Zehnder interferometer (MZI) for pressure measurement was proposed. The working principle of the pressure sensor was explained in Figure 13c. When light propagated in the fiber core, it would be divided into two main parts: one through the air cavity and the other in the fiber core. The two parts recombined at the output end of the air chamber. First, a micro square structure of about a 24 μm side length on the SMF end facet was made. Then, the fiber head with the microstructure was spliced together with another SMF head with a splicer to form a hollow ball. A microchannel at the top of the gas chamber perpendicular to the fiber axis was fabricated by using the fiber laser micromachining technology. The structure allowed the high pressure gas to enter the gas chamber. The sensitivity of 8239 pm /MPa was obtained under the pressure of 0–8 bar [153].

A miniature sensor for accurate measurement of pressure and depth with temperature compensation in the ocean environment was described. The sensor was based on an optical fiber extrinsic F-P interferometer (EFPI) combined with a fiber Bragg grating (FBG) [154]. The EFPI pressure sensor was fabricated by splicing the polished capillary and a multimode fiber (with a diameter of 200 μm) to form a diaphragm. The SMF with the fiber Bragg gratings was cleaved and inserted into the capillary. The combined pressure and temperature sensor system was mounted on-board with a mini remotely operated underwater vehicle in order to monitor the pressure changes at various depths. From the structure of this extrinsic F-P interferometer pressure sensor (Figure 13d), light propagated in single mode fibers and was reflected at the glass–air interface of the fibers, the air–glass interface, and the glass–water interface of the diaphragm. E_0_ was the light reflected at the end-face of the fiber. E_1_ was the light reflected at the inner side of the diaphragm, and E_2_ was the light reflected at the outer side of the diaphragm. The reflected light traveled back and formed interference with the other. FBG provided temperature measurements. The reflected optical spectrum from the sensor was monitored online, and pressure or temperature change caused a corresponding observable shift in the received optical spectrum. The measurements illustrated that the EFPI-FBG sensor could reach an accurate depth of ~0.020 m.

To be applied in high temperature environments, the pressure sensors utilizing SOI were presented. The thickness and the width of the sensitive diaphragm were 30 μm and 1000 μm, respectively. At room temperature and high temperature, both of the nonlinear errors were below 0.1%, and the hysteresis was less than 0.5%. High temperature measurements for the SOI pressure sensor showed that it could be used in a harsh environment with temperatures up to 350 °C [155]. By improving the thermal stability of ohmic contacts, the high work temperature of the pressure sensor could reach up to 500 °C [156]. The nonlinearity error of 0.17% FS and the sensitivity of 0.24 mV/kPa were obtained in the pressure range of 30–150 kPa as it worked at 500 °C, indicating that there was a good thermal stability of the ohmic contacts for the prepared pressure sensor.

In the application market of intelligent air conditioning control and home automation systems, there was a demand for pressure sensors that could withstand high voltage pulse. A high performance differential pressure sensor was reported, formed with a thin film under bulk micromachining technology [157]. The micro-opening inter-etch and sealing (MIS) processes were used to produce piezoresistive sensors similar to [158,159]. The conformal polySi layer was deposited by LPCVD in micropores on the inner wall of the cavity. The film was formed by polySi, where a single crystalline silicon structure with a beam island shape formed after etching. The chip size was 1.2 mm × 1.2 mm, and the sensitivity of 3.66 mV/V/kPa and linearity of 0.1% in full scale were observed. Based on the same process, focused on reducing chip size, a 0.6 mm × 0.6 mm × 0.45 mm sensor chip was fabricated, with sensitivity and nonlinearity of 0.029 mV/V/kPa and ±0.09% FS between 100–750 kPa [160]. To further reduce the chip size to 0.4 mm × 0.4 mm, a 2~3 μm thick bulk Si thin film was prepared by DRIE [161]. The sensor showed a good sensitivity of about 0.30 mV/V/kPa and a good nonlinearity of ±0.32%, within the pressure range of 20–100 kPa. Due to the good performance, the pressure sensor could be used in a barometer.

It is well known that tire failure might result in vehicle accidents. The miniaturized pressure sensor could monitor the change of tire pressure and transmit the data to the electronic information system [162,163]. The MIS process was also reported to fabricate piezoresistive pressure sensors for tire pressure monitoring system (TPMS). Deep grooves were etched out by DRIE through the small holes. Then, Tetramethylammonium hydroxide (TMAH) etchant was used to complete the inter-release by lateral under etching along the (110) and (211) orientations, resulting in a cavity [159,164]. The deposited polySi refilled the narrow release holes to seal the cavity. Based on this technology, a composite sensor integrated with a pressure sensor and accelerometer was made [165]. The composite sensor had a small size of 1.25 mm × 1.25 mm × 0.45 mm and a pressure sensitivity of 0.03 mV/V/kPa within the pressure range of 100–500 kPa. Other tiny wireless pressure sensor suitable for TPMS were reported [166,167,168]. The volume of a single chip was also sub mm^3^, and over the pressure of 0–1 MPa, there was still good linearity, below 0.01% FS.

In order to save the power supply of the TPMS system, the pressure sensor and piezoresistive accelerometer could be integrated on the same microchip by using surface micromachining technology and the piezoresistive effect [162]. Four polySi piezoresistors were placed in corresponding maximum stress positions. Piezoresistive accelerometers with clamped beam mass structures were designed, and piezoresistors were placed in four suspended beams. In order to obtain high quality and fast speed, the patterned film was electroplated with copper. Low temperature oxide and PSG were used as sacrificial layers, on which S_3_iN_4_ thin films were deposited to be the film, and an 8 μm thick copper layer was electroplated to fabricate accelerometers. Finally, the wafer level packaging was completed, and the beam mass structure was enclosed in a cavity. An SEM image of the TPMS composite sensor chip and the amplification part of the pressure sensor part are shown in Figure 14a. The Cu mass and spring beam consisted of the accelerometer, while the piezoresistors and Si_3_N_4_ rectangular diaphragm consisted of the pressure sensor. The size of the pre-packaged sensor was 1.6 mm × 1.6 mm × 0.9 mm. The test results showed that the sensitivity of the pressure sensor was 27.8 mV/MPa/V and the linearity of FSO was 0.34% in the range of 0–450 kPa. The sensitivity of the accelerometer was 5.2 μV/g/V in the range of 125 g.

Different from the common fiber optic sensors based on Bragg gratings or F-P, a microfiber optical pressure sensor made (50 μm × 130 μm × 130 μm) by direct laser writing technology (DLW) [169] was reported [170]. A drop of IP-DIP (dual in-line package) photoresist was dripped onto the cover slide in the DLW microscope room, then the optical fiber with a plane was installed in the center of the field of view of the microscope, just below the cover slide, and then, direct laser writing was carried out. The sensor consisted of a series of thin plates supported by springs that compressed under the applied force. The sensor was mounted on top of the fiber and coupled to the fiber by white light illumination, which was partially reflected and partially transmitted through each polymer plate. Figure 14b shows the schematic structure of this optical pressure sensor. The sensor included three 1.5 μm thick polymer plates, which were suspended at 13 μm above the output surface of the single mode fiber by means of four springs. There was a thicker (10 μm) fourth plate that was also attached to the spring and used as a pad. When force was applied, the spring compressed and brought the polymer plates closer together, which supported the force (or compression) sensing capability. The near-end spectrometer was used to monitor the change of reflection spectrum, reflecting the change of sensor compression. In the range of 0–50 N, the measurement error was about 1.5 N. These devices had potential application promise in the fields of velocity monitoring and in vivo medical imaging.

### 3.4. Consumer Electronics

The MEMS microphone [171,172] was an indispensable pressure sensor in mobile phones and the miniaturization and high sensitivity played a positive role in improving the ratio of performance/price [173]. Sandia National Laboratories reported a dual backplate condenser microphone, with Sandia’s Ultra-planar Multi-level MEMS Technology (SUMMiT). SUMMiT combined advanced IC process techniques with micromechanical fabrication to create complex micro-assemblies [174,175]. Through depositing multilayer polySi and the release process, the capacitive microphone with a 230 μm radius and 2.25 μm thick circular diaphragm was fabricated. The doped polySi layer acted as the upper and lower plates of the capacitor. A sensitivity of 282 μV/Pa and a linear response up to 160 dB at 1 kHz were obtained.

For unmanned aerial vehicles (UAV) or GPS, MEMS pressure sensors could be used as altimeters. Bosch introduced a miniature barometric sensor fabricated by APSM technology, BMP 380, with a package size of only 2.0 mm × 2.0 mm × 0.75 mm. Experiments in real life confirmed that the relative accuracy of the sensor was ±0.06 hPa at 25 to 40 °C, and the absolute accuracy was ±0.5 hPa at 0 to 65 °C in the absolute pressure range of 300 to 1100 hPa. ST Microelectronics reported a subminiature piezoresistive absolute pressure sensor LPS22HH. The sensing element for detecting absolute pressure was composed of a suspended film made by a special process developed by ST Microelectronics as mentioned above. The package size was 2.0 mm × 2.0 mm × 0.73 mm. An absolute pressure accuracy of 0.5 hPa could be obtained between 40 and 85 °C in the absolute pressure range of 260 to 1260 hPa.

In addition, with the popularization of intelligent terminals, wearable electronic devices had broad prospects in the consumer electronics market. As one of the core components, flexible wearable pressure sensors had the characteristics of being light, portable, excellent electrical performance, and high integration [176,177]. Common materials such as metals, inorganic semiconductors, and organic and carbon materials were taken as sensitive materials. By combining flexible substrates (such as PDMS) with these nanomaterials with good electrical properties, high performance wearable pressure electronic devices would be realized.

In order to show the parameters of the miniature pressure sensor more intuitively and more generally, we summarize the characteristics of some representative devices mentioned above and list them in Table 2.

## 4. The Condition of Sensor Size Reduction

Throughout the development history of pressure sensor miniaturization, excellent performance often required a large diaphragm area, which lead to mutual limitations between chip miniaturization and performance. In general, to apply miniature pressure sensors to various fields, a compromise between the two aspects should be balanced [178].

### 4.1. Problems Faced

Taking thin film as an example, based on the plate and shell theory [179], if a certain proportion of Length/Thickness (L/T) was maintained with the decrease of size L, the thickness T of thin film would be small, which led to a degeneration in linearity. Meanwhile, the small size generated carrier noise [180] and mechanical instability [181] that could not be ignored. Meanwhile, the stress average effect would be dominated [182], which worsened the performance of pressure sensors. The equivalent noise pressure from piezoresistive pressure sensors increased as 1/r^4^, where r was the equivalent radius of the diaphragm [183]. If the diaphragm was less than 50 μm in diameter, the noise pressure from Brownian motion would reach several mmHg [184]. On the other hand, from the perspective of semiconductor processing technologies, photolithography and etching processes had a great impact on the critical dimensions [185]. For example, for the production of resistance strips, with decreasing the size of the diaphragm, the resistance size decreased accordingly. At this time, the process error caused by the two processes would result in the non-negligible mismatching of resistance.

In addition, the structure layout, such as the electrode layout, was also a reason for the limited size. When the film was reduced to 100 μm, the size of the electrode was almost the same. At this time, the choice of the bonding pads became the determinant of the overall size of the chip. Different arrangements resulted in different sizes in the transverse dimensions of sensors with the same diaphragm [186].

### 4.2. The Solution

Fortunately, researchers found some ways to solve these problems and produced a batch of sensors with small sizes and good performance. For the compromise between small size and performance in the pressure sensor, the beam island structure on the diaphragm was adopted, and excellent performance was achieved with a very small size of the diaphragm [187,188]. The presence of the beam island provided a stress concentration area for the diaphragm where piezoresistors should be placed for great pressure sensitivity.

In addition to the beam island structure, since the discovery of the giant piezoresistance behaviors of Si/Al/Ag/SiC nanowires [189,190,191], there were many reports [192,193] about them. A Nano-Electromechanical System (NEMS) pressure sensor with high linearity and sensitivity was reported [194]. Photolithography and ion implantation were performed on the device layer of SOI. After etching the top silicon, SiNWs were formed with a cross-section of 100 nm × 100 nm and a 1 μm length. SiO_2_ and Si_3_N_4_ were then deposited for passivation. The sensor with a 1.3 μm Si_3_N_4_ layer had a good sensitivity of 4 mV/V/psi. Smaller silicon nanowires (90 nm × 90 nm) were developed to be sensitive elements on the basis of 200 μm square diaphragms. The pressure sensor had the advantages of a low initial center deflection of 0.1 μm and a high sensitivity of 6 mV/V/psi and a good linearity in the range of 0–20 psi [195].

The piezoresistive effect of then-type 3C-SiC nanowires under different loading forces was studied [196]. The pressure sensor based on the 3C-SiC nanowire had a high sensitivity, and the current at the pA level could be obtained by applying forces at the nN level. The transverse piezoresistive coefficient π [110] of the nanowires ranged from 0.75 to 7.7 × 10^−11^/Pa when the applied force was between 25.59 and 153.56 nN. In recent years, silver nanowires (AgNWs) were gradually applied to the conductive network of flexible piezoresistive materials due to their advantages of high conductivity and flexibility [197]. By a conductive cotton piece with AgNWs (10^−4^–10^−5^ Ω·cm) for hydrogen bonding, a AgNW conductive network was formed [191]. The bio based flexible pressure sensor based on the conductive network had a high sensitivity (3.4 kPa^−1^) and a fast response and relaxation time (<50 ms), and the pressure sensor could be widely used in speech recognition and robot systems.

Electrodes were not only used as part of a conductive circuit in previous works, but also a new sensing method based on pulse micro-discharge [198,199]. This method combined the electrode with the sensing unit and measured pressure change by the distribution current between two electrodes [200]. This new structure had a small size as illustrated by the following examples, and at the same time, its working principle is briefly described below.

A micro-discharge based pressure sensor was fabricated by using through-wafer isolated bulk silicon lead transfer. First, etching low resistance p-type silicon substrate to create three deep grooves by DRIE and filling the grooves with silica to form the electrical isolation of the three parts were performed. The base silicon was then thinned by CMP to reach the deep groove. After that, metal schematization was carried out on the upper and lower surfaces corresponding to the three isolated parts, which were A2/A1/K pads, respectively. Then, 3 μm thick amorphous silicon was deposited on the surface, followed by depositing Ni/Al on it, and metal stripping was done to make it only connected with A2. The oxide-nitride-oxide was then deposited, serving as the diaphragm. Then, the release holes were etched, followed by releasing the amorphous silicon with XeF_2_. Finally, the Plasma Enhanced Chemical Vapor Deposition (PECVD) aluminum oxide was deposited to seal the holes and cavity. In addition, the sensor also had a channel connected to the chamber for filling discharge gas Ar. When a high voltage of 100 V was applied between the cathode and the anode, the electric breakdown occurred in the cavity and resulted in discharge and current I1 and I2. Applying pressure, the diaphragm deflected and the electrode spacing changed between K and A2, while the spacing between K and A1 remained constant. This process redistributed the ionized species in the cavity and led to a change in terminal currents [201], while the change in pressure was converted by a change in differential current (I1 − I2)/(I1 + I2). The size of the device was 300 μm × 300 μm ×150 μm. As the external pressure increased from 1 atm to 8 atm, the (I1 − I2)/(I1 + I2) monotonically increased from −0.7 to 0.2. Other micro-discharge based pressure sensors were fabricated [202,203]. A discharge based pressure sensor for high temperature applications was introduced. The diameter of the diaphragm was 1 mm, and the sensor was 125 μm in thickness [204]. A high-pressure sensor utilizing micro-discharge provided a sensitivity of 42,113 ppm/atm under 15 atm. The 25 μm thick nickel diaphragms had diameters of 800 μm, and the device was enclosed within ceramic packages of 2050 μm × 1650 μm [205]. The size continued to go down, and it went down to 0.057 mm^3^ for the active volume in [206]. The structure was encapsulated in a 1.6 mm^3^ ceramic surface mounting package.

In addition to the new structure and new layout, researchers made considerable progress on the exploration of new materials to bring the sensor size and performance into a win-win situation. Single walled carbon nanotubes (SWCNT) were expected to be used as sensor elements, providing ideas for the conversion of miniature MEMS to NEMS [207,208]. The works, using SWCNT as piezoresistors, were also performed [209,210,211]. A piezoresistive pressure sensor with parallel integration of individual SWCNT was reported. The process began with the thermal oxidation on a silicon substrate. Pd was then deposited on it and patterned to serve as an electrode. The device was immersed in an aqueous dispersion for selective dielectrophoretic of SWCNT. Then, SWCNT was selectively deposited onto the prefabricated electrode by dielectric electrophoresis (DEP) [212,213] assembly. A layer of Al_2_O_3_ was deposited and etched for protection. The cavity was etched out with back alignment and the SWCNT located on the membrane edges (Figure 15a). The SWCNT were arranged radially on the edge of the circular membrane. When applying pressure on the film, the current flowing through SWCNT would change accordingly. Figure 15b gives the size of the membrane, a diameter of 100–120 μm, and a thickness of 190 nm with SiO_2_/Al/Au. SWCNT were encapsulated between 70 nm SiO_2_ and 80 nm Al_2_O_3_ layers. The sensor had a sensitivity of 0.25 ΔR/R/bar, and the resolution was better than 50 mbar. The power consumption was less than 40 nW. Similarly, using SWCNT field-effect transistors (FETs) as strain gauges, an ultra-small pressure sensor was reported [214,215]. The process was similar to the methods in [216,217] and used methane as the carbon feedstock. The individual SWNTs were synthesized on a highly doped silicon substrate with a 200 nm thick thermally grown silicon oxide layer from ferritin based iron catalyst nanoparticles in CVD. The remaining steps were similar to traditional MOSFET. The minimum piezoresistive pressure sensor with a diameter of 40 μm film electric parameter was developed. Under the pressure range of 0–200 mbar, the sensitivity was 54 pA/mbar (V_ds_ = 200 mV).

Besides SWCNT, graphene was also an interesting material for NEMS because of its thin thickness, high carrier mobility, and high Young’s modulus [218,219,220,221,222,223]. The method of transferring graphene to seal the cavity was also studied [224,225,226,227,228].

A polymethyl methacrylate (PMMA) or polycarbonate (PC) layer was covered on a side of the graphene to act as a mediator with the final substrate. Graphene was etched with O_2_ plasma on the back of the foil, and then, copper foil was etched away with wet FeCl_3_. PMMA/graphene films were picked up and dried on a hot plate by slicing. The polymer layer was etched after placing graphene on the designed substrate. The photoresist layer was applied and exposed to form graphene patterns. The desired shape of graphene was etched using O_2_ plasma etching [229,230]. The diameter of the graphene film was 18 μm. Figure 16a shows an SEM of a fabricated graphene pressure sensor. The graphene is shaded blue, the cavity green, the electrode and contact pad yellow, and the bonding line orange. When the graphene film was deformed by pressure, the resistance between two the pads would change. The 0.59% changes of the resistance on graphene films were measured in the pressure range from 523 to 1000 mbar.

Based on the pressure dependence of the resonant frequency of the film, several layers of graphene film were used as the pressure sensor of the extrusion film [231]. Graphene flakes were transferred on a dumbbell shaped hole in a SiO_2_ substrate (Figure 16b). The graphene flakes were transferred to a dumbbell shaped cavity on the SiO_2_ substrate, covering half of the dumbbell with the channel. The drum had a diameter of 5 μm and an oxide thickness of 400 nm. The different ambient gas pressures could change the stiffness of the resonator. The relationship between the resonant frequency of the membrane and the pressure was measured with the 4 MHz frequency moving between 8 mbar and 1000 mbar. The sensitivity of the pressure sensor was 9000 Hz/mbar, which was 45 times higher than the latest MEMS based extrusion film pressure sensor, while using 25 times less film area. Besides, a graphene pressure sensor with a Si_3_N_4_ square membrane was reported. The Si_3_N_4_ membrane had a thickness of 100 nm and a width of 280 μm. In a dynamic pressure range from 0 mbar to 700 mbar, a gauge factor of 1.6 could be observed [232].

Graphene could be combined with a polymer to form a capacitive pressure sensor. A membrane of 30 μm in diameter was fabricated. The pressure sensor gave an unprecedented pressure sensitivity of 123 aF/Pa/mm^2^ over a pressure scale of 0–80 kPa [233]. Graphene could also be used as a sensitive diaphragm in an F-P cavity to improve the sensitivity of pressure sensors [234,235]. An F-P interferometer was developed by the fabrication of 13 layers of graphene with a diameter of 125 μm [236]. According to the refractive index characteristics of the film, the influence of the graphene film layer and incident light angle on the reflectivity of the film was obtained. In the pressure range from 0 to 3.5 kPa, pressure induced deflection of 1096 nm/kPa and a sensitivity of 179 nm/kPa were achieved.

It is worth mentioning that flexible and scalable substrates could provide innovations in device structure, material selection, and manufacturing methods for pressure sensors. These flexible devices might be applied to personalized health monitoring, human–machine interfaces, and environmental sensing [76,237,238]. Although the overall sizes of the flexible devices were usually larger than 2 mm × 2 mm due to the existence of the substrate, their excellent ductility and exploration of the electrical properties of new materials could provide more possibilities for the flexible application in small sizes.

### 4.3. Possible Development Directions

Based on the above analysis, with the development and improvement of micro-machining technology and related theories of mechanics and crystallography, micro-pressure sensors are developing towards miniaturization. The possible development directions of small pressure sensors in the future are mainly focused on improved diaphragm structures, a new sensing element, a new layout structure, and a new material. The reduction of film size would result in the homogeneity of stress. Improved diaphragm structures, such as the addition of micro-beams and islands, generated stress concentrations that maintained high sensitivity while maintaining good nonlinearity. New materials and sensing elements, such as SWCNT, graphene, and SiC/Ag nanowires with their giant piezoresistive effects and excellent electrical transport properties, could provide sensitivity factors, equivalent or even superior to traditional large sized thin film pressure sensors at small scales. The new structure optimized the layout and increased the effective utilization of chip area. For example, a micro-discharge based pressure sensor combined the electrode with the sensing unit, realizing the effective utilization of electrodes.

## 5. Conclusions

Throughout the development history of miniature pressure sensors, the replacement of metal strain gauges with semiconductor piezoresistors accelerated the minimization of pressure sensors in the middle of the 20th Century. With the generation and update of semiconductor microfabrication technology, the size of the pressure sensors has been further shrinking. Recently, there was a huge demand for small pressure sensor applications in various fields. However, the worse performance and difficult fabrication procedures are inevitable problems on the road of minimization of pressure sensor. The possible approaches such as improved diaphragm structure, new sensing elements, new layout structures, and new materials, were proposed. The common development of decreasing size and improved performance of pressure sensors was demonstrated.

## Figures and Tables

**Figure 1 micromachines-11-00056-f001:**
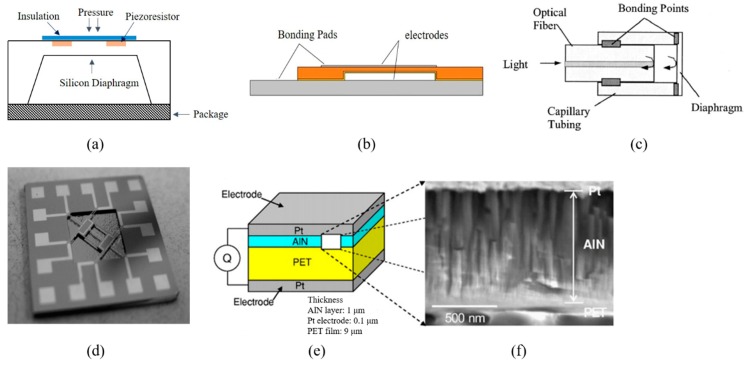
Five kinds of working principles for pressure sensors. (**a**) Typical structural schematic diagram of a piezoresistive absolute pressure sensor. The diaphragm would be subjected to pressure and be deformed. (**b**) Capacitive transducer with a metalized diaphragm over a conductive base of stainless steel foil. The diaphragm layer was a thin layer of polyimide, 5 μm. Reprinted with permission from [12]. Copyright 2016 Elsevier B.V. (**c**) Typical structural schematic diagram of an optic fiber pressure sensor. The device structure was all-silica, and the light experienced two reflections. Reprinted with permission from [17]. Copyright 2005 The Optical Society of America (OSA). (**d**) Optical image of a resonant pressure sensor with a suspended “H”-type silicon resonant beam. Reprinted with permission from [23]. Copyright 2009 The International Society for Optics and Photonics (SPIE). (**e**) Typical structural schematic diagram of a piezoelectric pressure sensor. The AlN layer converted pressure into the voltage change through the piezoelectric effect. (**f**) Scanning electron microscope (SEM) image of the cross-sectional structure of aluminum nitride (AlN) film deposited on polyester (PET) film. Reprinted with permission from [28]. Copyright 2006 American Institute of Physics (AIP).

**Figure 2 micromachines-11-00056-f002:**
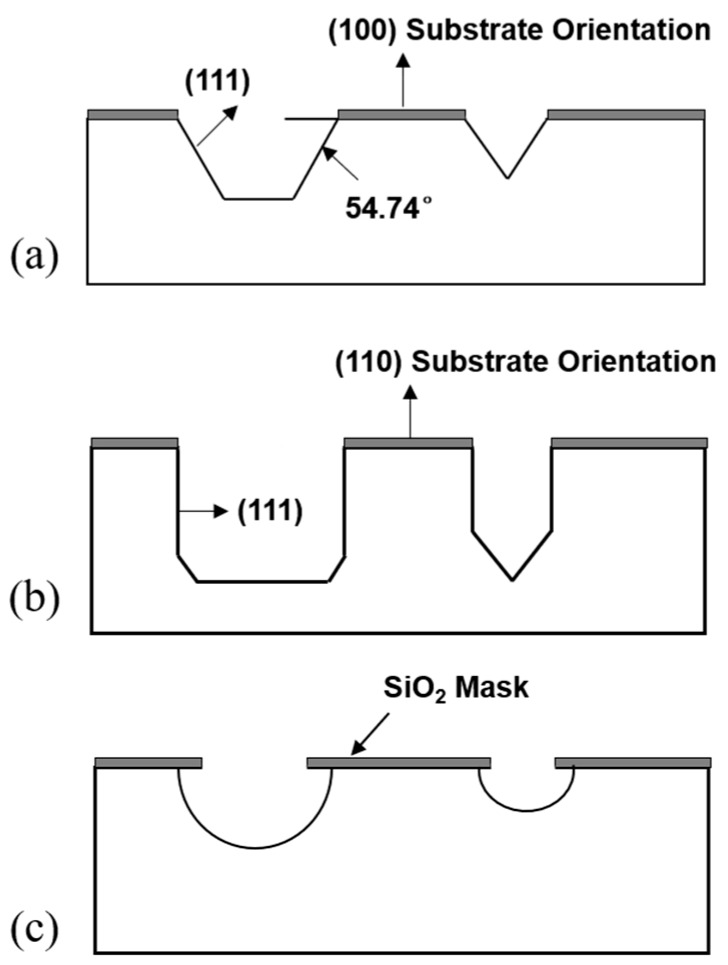
Different corrosion processes of silicon in the fabrication of cavities. (**a**,**b**) Anisotropic etchants had different etching rates for different crystal planes and could produce the cavities with vertical sidewalls or slopes. (**c**) Semi-circular grooves with transverse expansion could be fabricated by the isotropic corrosion solution.

**Figure 3 micromachines-11-00056-f003:**
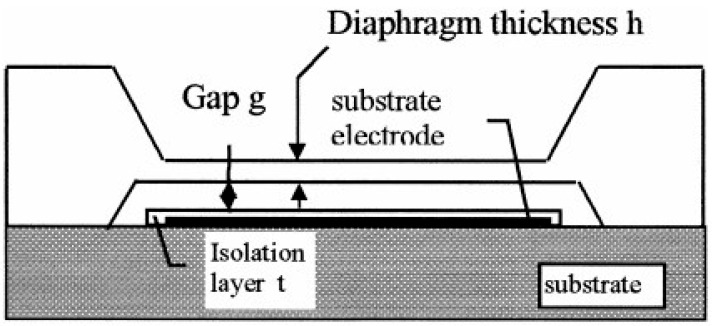
Cross-section diagram of a typical silicon-glass capacitive pressure sensor. The silicon diaphragm would be deformed by the applied pressure, which changed the distance of the gap. Reprinted with permission from [52]. Copyright 1999 Elsevier Science S.A.

**Figure 4 micromachines-11-00056-f004:**
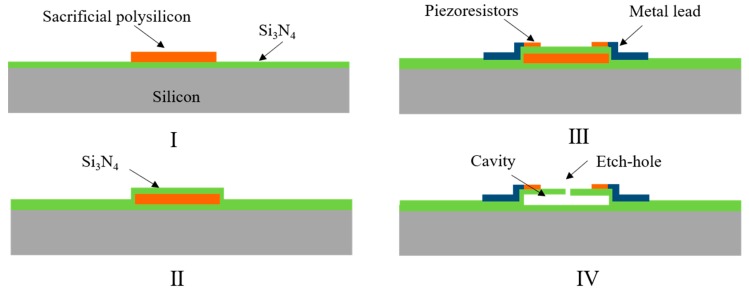
Pressure sequence of the surface micro-diaphragm pressure sensor. (**I**) depositing sacrificial polySi on silicon substrate, (**II**) depositing Si_3_N_4_ on the polySi layer to be the pressure sensitive diaphragm, (**III**) fabrication of piezoresistors and metal lead, and (**IV**) releasing the sacrificial layer through etch-holes to make a cavity.

**Figure 5 micromachines-11-00056-f005:**
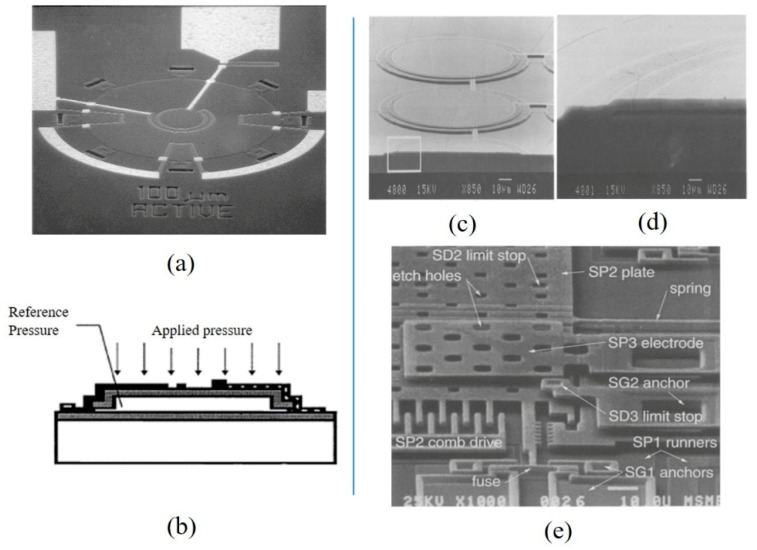
The membrane-cavity structures fabricated by surface micromachining technology. (**a**) SEM image of a piezoresistive pressure sensor. The Si_3_N_4_ diaphragm had a diameter of 100 um, with six radial and one circumferential piezoresistors. (**b**) PolySi piezoresistors were formed at the top of the Si_3_N_4_ film. The sealed vacuum chamber under the film provided reference pressure. Reprinted with permission from [63]. Copyright 1995 SPIE. (**c**) Cross-section image of a capacitive pressure sensor. The circular pressure sensitive film was made of PolySi with diameters ranging from 50 to 120 μm. (**d**) The gap between the two electrodes was 900 nm. Reprinted with permission from [60]. Copyright 1994 Elsevier Science S.A. (**e**) SEM image of a planarized three polySi surface micromachining structure. With the surface microfabrication technology, a multi-layer suspended structure of the MEMS pressure sensor could be fabricated. Reprinted with permission from [62]. Copyright 1994 Springer Verlag. SD, stand-off; SG, sacrificial glass.

**Figure 6 micromachines-11-00056-f006:**
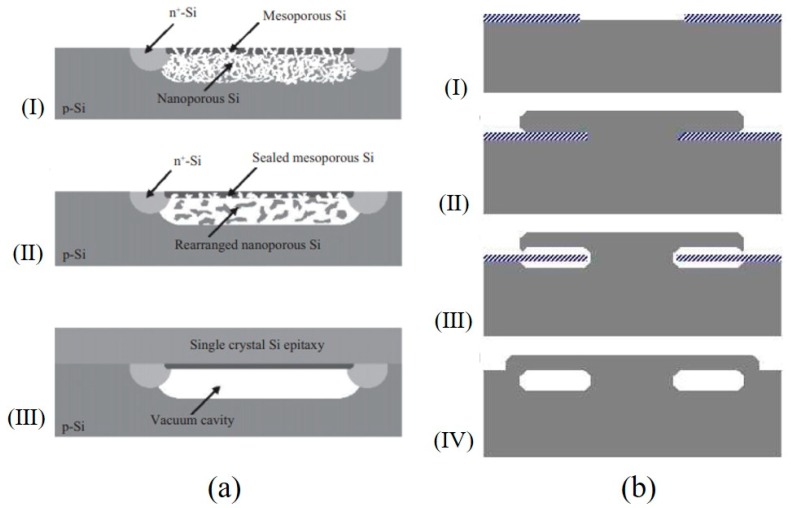
Utilizing surface migration at high temperature, the hollow cavity was made by Bosch and ST Microelectronics. (**a**) Advanced porous silicon membrane (APSM) process for pressure sensors developed by Bosch. The buried channel was fabricated by surface migration of porous silicon at high temperature. Reprinted with permission from [36]. Copyright 2018 CIE. (**b**) Fabrication process of the ST Microelectronics pressure sensor. The buried channel was fabricated by surface migration of SiO at high temperature. Reprinted with permission from [68]. Copyright 2006 Springer Verlag.

**Figure 7 micromachines-11-00056-f007:**
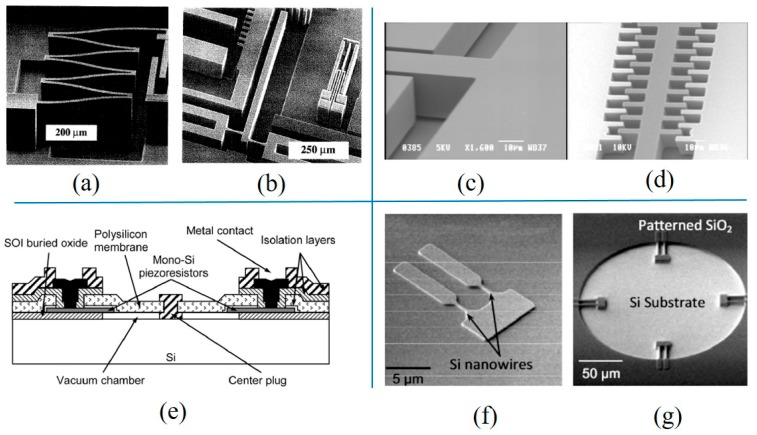
Miniature structures with small sensitive elements were fabricated with deep reactive ion etching and silicon on insulator (SOI). (**a**) SEM image of an electrostatic resonator. The spring fabricated by deep reactive ion etching had a steep sidewall. (**b**) SEM image of a thermal actuator. Reprinted with permission from [71]. Copyright 1996 Elsevier Science S.A. (**c**) Straight beams of a silicon capacitive pressure sensor, with sections of 20 μm × 30 μm. (**d**) Comb beams with a 2 μm critical dimension. Reprinted with permission from [73]. Copyright 2002 SPIE. (**e**) Cross-section of the schematic silicon on insulator (SOI) piezoresistive pressure sensor. The piezoresistors were fabricated by ion implantation into the device layer, and the cavity was released by etching the buried oxygen layer. Reprinted with permission from [74]. Copyright 2013 Elsevier B.V. (**f**) SEM image of a pair of patterned silicon nanowires on the SiO_2_ layer. Using reaction ion etching, silicon nanowires were fabricated, with a size of 10 μm × 100 nm × 100 nm. (**g**) Location of the silicon nanowires (SiNWs). Silicon nanowires were placed symmetrically at the edge of the film to obtain large stress changes. Reprinted with permission from [78]. Copyright 2010 Elsevier Ltd.

**Figure 8 micromachines-11-00056-f008:**
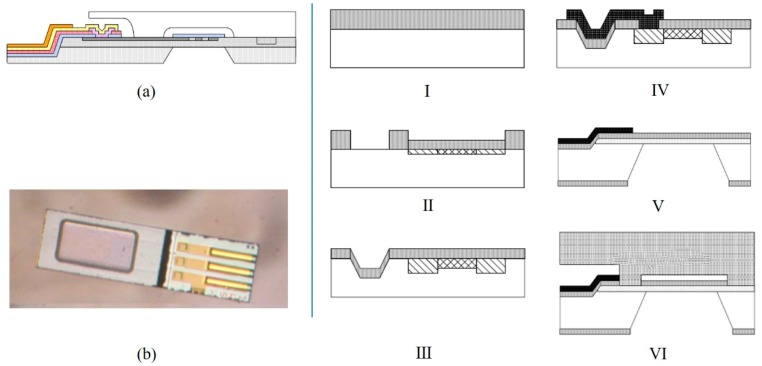
An absolute piezoresistive pressure sensor packaged by anodic bonding with the cavity and silicon wafer, used for the treatment of coronary artery stenosis. (**a**) Cross-section diagram of the silicon absolute piezoresistive pressure sensor. The cavity was made in the back of the silicon substrate, and the pressure sensor was packaged with grooved ground glass by alignment bonding. (**b**) Optical image of the top of the sensor die. (**I**–**VI**) Production process of the pressure sensor. Reprinted with permission from [89]. Copyright 2001 Materials Research Society (MRS).

**Figure 9 micromachines-11-00056-f009:**
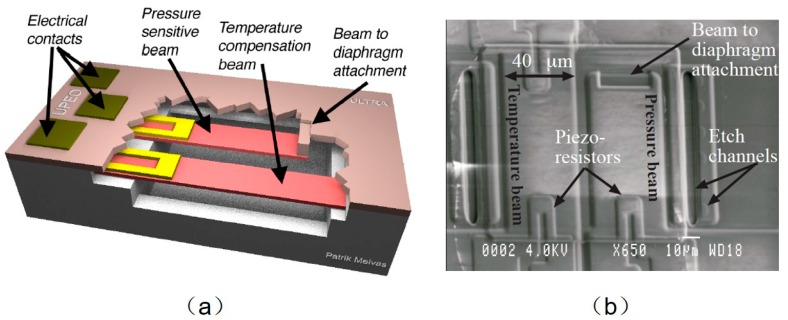
A temperature compensated strain gauge beam pressure sensor. (**a**) Schematic drawing of the polySi beam pressure sensor. (**b**) SEM image of the fabricated dual-beam pressure sensor. The two beams with a piezoresistor were suspended below the film through the attachment. Reprinted with permission from [90]. Copyright 2002 Elsevier Science B.V.

**Figure 10 micromachines-11-00056-f010:**
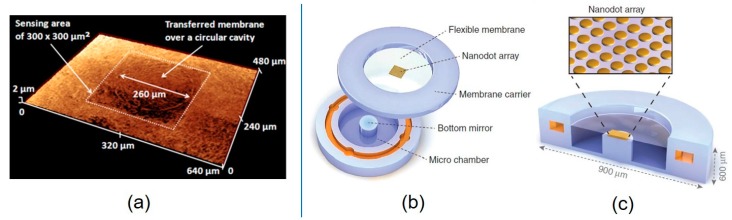
A capacitive pressure sensor and an optical pressure sensor applied for intraocular pressure measurement. (**a**) Surface profile of the capacitive pressure sensor by parylene bonding. Reprinted with permission from [115]. Copyright 2012 Springer. (**b**) Structural schematic diagram of the optical pressure sensor with the flexible silicon nitride membrane and bottom mirror. (**c**) Cross-section of the schematic assembled sensor and image of the nanodot array (inset). Reprinted with permission from [2]. Copyright 2017 Open Access.

**Figure 11 micromachines-11-00056-f011:**
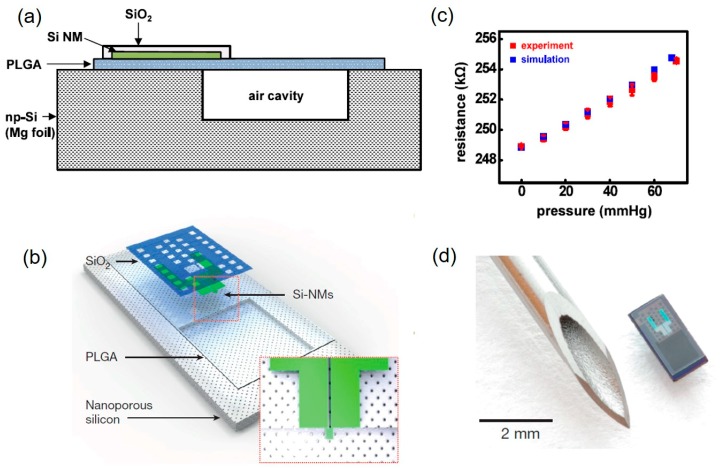
Bioresorbable silicon pressure sensors for the intraocular pressure measurement. (**a**) Cross-section diagram of the bioresorbable silicon pressure sensor. The porous silicon substrate with an air cavity was combined with PLGA (lactide: glycolide composition with 3:1 wt%) to form a reference pressure chamber. (**b**) Structural schematic diagram of the biodegradable pressure sensor. The Si-nanomembrane was processed into serpentine patterns as piezoresistors. (**c**) Variation of resistance with applied pressure, (**d**) Optical image of the pressure sensor. Reprinted with permission from [130]. Copyright 2016 Macmillan.

**Figure 12 micromachines-11-00056-f012:**
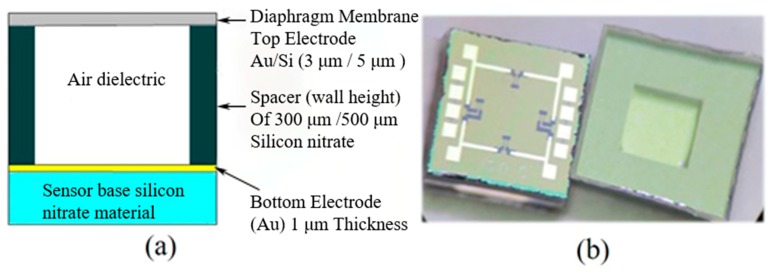
A capacitive pressure sensor for a spacecraft altimeter and a piezoresistive absolute pressure sensor for aerospace applications. (**a**) Cross-section diagram of the capacitive pressure sensor. The diaphragm could be made by gold, Si, and liquid crystal polymer (LCP). Reprinted with permission from [135]. Copyright 2012 Open Access. (**b**) Front side view and backside view of the piezoresistive absolute pressure sensor die after dicing. Reprinted with permission from [143]. Copyright 2014 Springer.

**Figure 13 micromachines-11-00056-f013:**
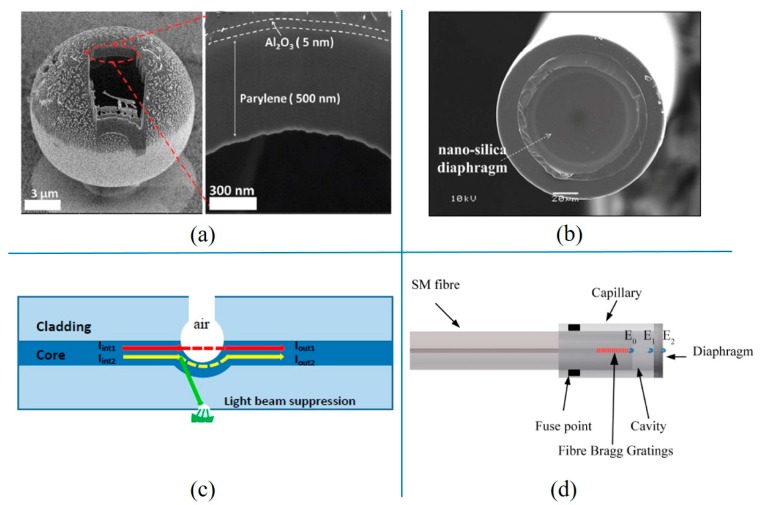
Optical pressure sensors used for industrial application. (**a**) SEM image of one micro-balloon and the image on the right showing a magnified image of the parylene shell with the Al_2_O_3_ diffusion barrier. Reprinted with permission from [151]. Copyright 2016 The Royal Society of Chemistry (RSC). (**b**) SEM image of the ultrathin silica diaphragm fused at the end of the silica capillary. Reprinted with permission from [152]. Copyright 2017 Open Access. (**c**) Structural schematic diagram of the light propagation in the fiber in-line Mach–Zehnder interferometer. Part of the light propagating in the fiber passed through the air cavity, and the other passed through the core. Reprinted with permission from [150]. Copyright 2015 SPIE. (**d**) Structural schematic diagram of the extrinsic Fabry–Perot interferometer sensor. Reprinted with permission from [154]. Copyright 2017 Open Access.

**Figure 14 micromachines-11-00056-f014:**
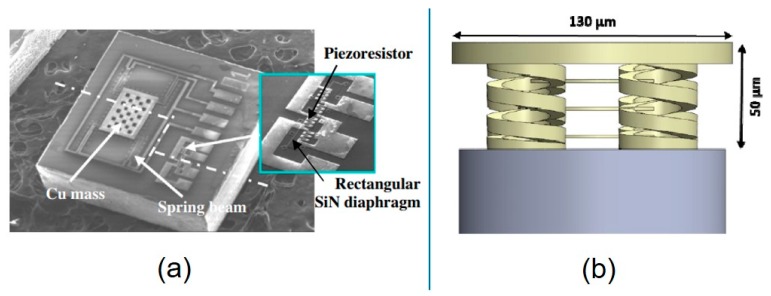
An integrated electronic device with a piezoresistive pressure sensor and a piezoresistive accelerometer applied for tire pressure monitoring system. A microfiber optic force sensor fabricated by direct laser writing. (**a**) SEM image of the chip, the inset for the magnified pressure sensor. Reprinted with permission from [162]. Copyright 2011 Elsevier B.V. (**b**) Structural schematic diagram of the fiber optic force sensor. Reprinted with permission from [170]. Copyright 2018 The Optical Society of America (OSA).

**Figure 15 micromachines-11-00056-f015:**
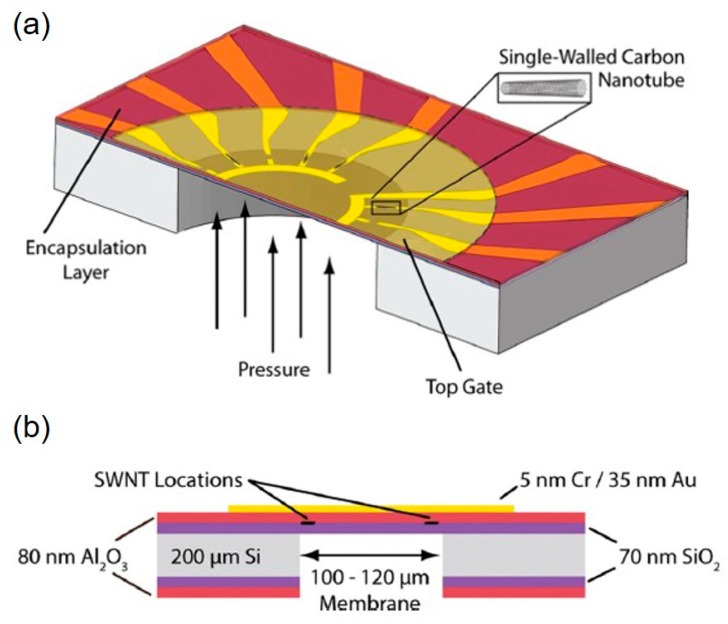
A piezoresistive pressure sensor with parallel integration of single walled carbon nanotubes (SWCNT). (**a**) Structural schematic diagram of SWCNT based pressure sensor. SWCNT were arranged along the edge of the circular film radially, and an encapsulation layer of alumina was deposited to protect them. (**b**) Cross-section diagram of the chip and membrane layer architecture. Reprinted with permission from [213]. Copyright 2011 American Institute of Physics (AIP).

**Figure 16 micromachines-11-00056-f016:**
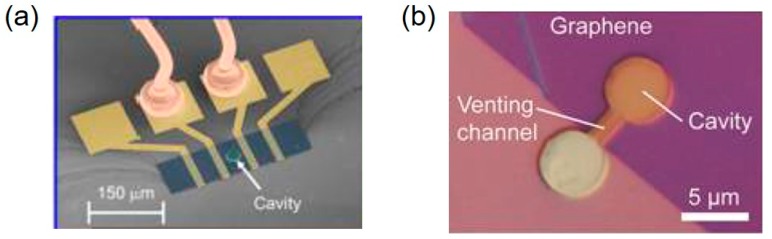
Graphene based piezoresistive pressure sensors. (**a**) SEM image of a piezoresistive pressure sensor with a circular graphene membrane. Reprinted with permission from [229]. Copyright 2016 ACS. (**b**) Half of the graphene flake dumbbell was covered, thus creating a drum with a venting channel. Reprinted with permission from [231]. Copyright 2015 ACS.

**Table 1 micromachines-11-00056-t001:** Pressure sensors using different materials and bonding techniques.

Diaphragm Material	Substrate Material	Assembly Technology
Single crystal silicon	Glass	Anodic bonding
Single crystal silicon	Silicon	Fusion bonding
Polysilicon	Silicon	Surface micromachining
Silicon nitride	Silicon	Surface micromachining
Polymeric materials	Silicon	Surface micromachining
Metal	Glass/ceramic	Eutectic bonding, soldering
Metal	Polymer	Polymeric seals
Ceramic(metalized)	Ceramic	Glass seal, metal seal
Polymeric materials (metalized)	Polymers	Polymeric seals, glue

**Table 2 micromachines-11-00056-t002:** Characteristics of some typical referenced pressure sensors. Intraocular pressure (IOP), interocular pressure (“CAP” represents “capacitive”; “PZR” represents “piezoresistive”).

Micromachining Methods	Transduction Mechanisms	Characteristic Dimensions (µm)	Pressure Range	Application	Reference
Bulk silicon process	CAP (Si)	1500 × 547 × 5	0–140 psi	Industrial	[52]
Surface silicon process	PZR (Si_3_N_4_)	100 × 100 × 0.8	0–300 kPa	Ultrasonic	[56,59]
	CAP (PolySi)	120 × 120 × 1.5	0–10 bar		[60]
	PZR (PolySi)	103 × 103 × 0.4	−3–40 kPa	Blood	[88]
	PZR (Si)	280 × 130 × 5	−40–66 kPa	Blood	[89]
Fiber laser micromachining	PZR (Si)	550 × 550 × 4	0–10 kPa	IOP	[112]
	Optical (Si_3_N_4_)	π600^2^ × 0.3	0–7 kPa	IOP	[2]
	PZR (Si)	100 × 100 × 2	0–2 kPa	ICP	[122]
	CAP (SiC)	200 × 200 × 0.5	0.2–3.5 atm	Gas turbine	[142]
	PZR (Si)	750 × 750 × 210	0–400 bar	Aerospace	[144]
	Optical (parylene)	4/3π12^3 ^ × 0.4	0–20 Psi	Imaging	[151]
	Optical (fiber)	24 × 24	0–8 bar		[153]
	Optical (silica)	200 × 200 × 4	0–5 bar	Ocean	[154]
	Optical (silica)	130 × 130 × 50	0–10 N	Imaging	[170]
DRIE with SOI	PZR (PolySi)	π80^2 ^ × 1.2	0–100 kPa	Size demanding	[74]
	PZR (SiO_2_)	200 × 200 × 3.5	0–40 kPa	Medical	[78,79]

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
