# Peer review of "Recent Progress of Miniature MEMS Pressure Sensors"

_micromachines, 2020, doi:10.3390/mi11010056_

Round 1
Reviewer 1 Report
The submitted manuscript “Recent progress of miniature MEMS Pressure sensors” P. Song, et.al. is a review on miniaturization of pressure sensors which has been pursued for the last 50-60 years. The review introduces the readers with pressure sensors by describing the individual transduction mechanisms, and then it focuses on the actual miniaturization. This miniaturization is given within the historical frame and development of silicon-based manufacturing methods. A very interesting section is the section on possible applications of miniaturized pressure sensors especially in the medical field. Sensors used in industrial applications or in electronics are also described. The authors finish the review by describing the challenges for further sensor miniaturization (noise caused by Brownian motion or mechanical instability) and offer possible solutions.
Comments:
The quality of some of the figures is really poor. I understand that the figures were copied from other papers but they authors obtained permission for their use so I believe they could obtain figures of higher quality. The English is satisfactory but still I would recommend checking the English by a native speaker. Some sentences are a bit cumbersome (especially in the abstract and the introduction). There are misspellings throughout the manuscript (e. g. line 47 motioned instead of mentioned). The whole review is very descriptive and it is written in the style that someone did something. Each paragraph is usually devoted to a single publication. Could the authors summarize the review by creating one or two tables overviewing the referenced publications? This table could contain characteristic dimensions, transduction mechanisms (used membranes, electrodes,…), fabrication method, pressure range, application,…. Equation on line 63 is not correct (it misses permitivity). Line 22 – units are probably wrong.Author Response
Thanks for the reviewer’s careful reading of our manuscript. It’s very valuable and helpful for revising and improving our paper.
Please see the attachment!

Reviewer 2 Report
The English needs be improved before publication. Especially, many paragraphs and sentences lack appropriate transitions.
Some paragraphs needs to be separated. For instance, the paragraph starting at line 678 describes two types of sensors. It is better to separate it into two paragraphs. Or a proper transition sentence should be inserted in between.
A lot of details of microfabrication processes are present without figures. it would be very difficult for readers to understand. It's better to make these descriptions more concise.
Some reviewed sensors are accompanied with figures, but others are not.
There are many typos and grammar errors. for example, line 43: could perceived; line 47: motioned.
Line 63: the equation C=S/D is incomplete. Dielectric constant and vacuum permittivity are missing.
Line 425: 1.6 mm or um?
Line 578: what is Sie elastomer?
Starting on line 568, should a different section be used (e.g., biocompatibility issue)?
For the design shown in Fig. 11(a), it is surprising that the distance between two electrodes is 300 um or 500 um. this will lead to very small capacitance. Please double check.
Line 826: the capital T should be deleted.
Author Response
Thanks for the reviewer’s careful reading of our manuscript. It’s very valuable and helpful for revising and improving our paper.
Please see the attachment!
